# Role of Virus-Induced EGFR Trafficking in Proviral Functions

**DOI:** 10.3390/biom13121766

**Published:** 2023-12-09

**Authors:** Se Sil Noh, Hye Jin Shin

**Affiliations:** 1Department of Microbiology, Chungnam National University School of Medicine, Daejeon 35015, Republic of Korea; wotuz0123@o.cnu.ac.kr; 2Department of Medical Science, Chungnam National University School of Medicine, Daejeon 35015, Republic of Korea; 3Brain Korea 21 FOUR Project for Medical Science, Chungnam National University, Daejeon 34134, Republic of Korea; 4Research Institute for Medical Sciences, College of Medicine, Chungnam National University, Daejeon 34134, Republic of Korea

**Keywords:** epidermal growth factor receptor, EGFR inhibitors, intracellular trafficking, antiviral strategies, infectious diseases

## Abstract

Since its discovery in the early 1980s, the epidermal growth factor receptor (EGFR) has emerged as a pivotal and multifaceted player in elucidating the intricate mechanisms underlying various human diseases and their associations with cell survival, proliferation, and cellular homeostasis. Recent advancements in research have underscored the profound and multifaceted role of EGFR in viral infections, highlighting its involvement in viral entry, replication, and the subversion of host immune responses. In this regard, the importance of EGFR trafficking has also been highlighted in recent studies. The dynamic relocation of EGFR to diverse intracellular organelles, including endosomes, lysosomes, mitochondria, and even the nucleus, is a central feature of its functionality in diverse contexts. This dynamic intracellular trafficking is not merely a passive process but an orchestrated symphony, facilitating EGFR involvement in various cellular pathways and interactions with viral components. Furthermore, EGFR, which is initially anchored on the plasma membrane, serves as a linchpin orchestrating viral entry processes, a crucial early step in the viral life cycle. The role of EGFR in this context is highly context-dependent and varies among viruses. Here, we present a comprehensive summary of the current state of knowledge regarding the intricate interactions between EGFR and viruses. These interactions are fundamental for successful propagation of a wide array of viral species and affect viral pathogenesis and host responses. Understanding EGFR significance in both normal cellular processes and viral infections may not only help develop innovative antiviral therapies but also provide a deeper understanding of the intricate roles of EGFR signaling in infectious diseases.

## 1. Introduction

The multifaceted role of the epidermal growth factor receptor (EGFR) is of paramount importance not only in understanding human diseases in relation to cell survival, proliferation, and cellular homeostasis [1,2,3] but also in the intricate realm of viral infections. In a recent study, most evidence supported the roles of EGFR in facilitating viral entry, replication, or escape from the host immune response in viral infections [4].

Viral infections begin with the binding of the viral particles to specific cellular receptors, setting in motion a complex cascade of intracellular events meticulously crafted to manipulate the host cell machinery, thereby facilitating viral replication [5]. The selection of these receptors plays a pivotal role in guiding viruses to their desired tissues and orchestrating successful traversal of cellular barriers, allowing the viral genome to infiltrate host cells and propagate with remarkable precision [6]. Notably, viruses recognize and engage EGFR, a prominent cell surface receptor, as an indispensable component of their strategy for host cell entry [7]. The central role of EGFR in orchestrating viral entry, replication, and the evasion of immune responses has become increasingly evident. In this regard, EGFR trafficking plays a pivotal role in virus–host interactions by allowing the translocation of EGFR into an array of intracellular organelles, including endosomes, lysosomes, mitochondria, and even the nucleus [8].

This review discusses the recent progress in understanding the pivotal role of EGFR in viral entry and its overarching function as a coordinator that governs viral replication. We summarize the current state of knowledge regarding the intricate interactions between EGFR and various viruses, and the interactions that are unequivocally indispensable for the successful propagation of these viral agents. We have also curated data for a spectrum of viruses that were tested with EGFR inhibitors in vitro or in vivo and summarized them in Table 1. Exploring the function of EGFR in viral infections can offer insights into the development of innovative antiviral strategies.

## 2. Structure and Properties of the EGFR Protein

EGFR (ERBB1/HER1; 170 kDa) belongs to the human ErbB family of receptor tyrosine kinases (RTKs), which are transmembrane receptors with three functional domains: an extracellular domain (ECD), a transmembrane domain, and an intracellular domain (ICD). The ECD consists of two ligand-binding domains (L1 and L2) and two cysteine-rich domains (CR1 and CR2). The ligands of EGFR are epidermal growth factor (EGR), amphiregulin (AR), transforming growth factor alpha (TGF-α), epiregulin (EREG), heparin-binding epidermal growth factor (HB-EGF), betacellulin (BTC), and epigen (EPGN); the binding of these ligands with the receptor induces conformational changes to trigger signaling. The ICD has a conserved cytoplasmic catalytic tyrosine kinase domain and multiple tyrosine residues that contain two lobes, an adenosine triphosphate (ATP)-binding site and a substrate-catalysis site [1,42,43]. The intrinsic tyrosine kinase activity of EGFR induces substrate phosphorylation or autophosphorylation to trigger EGFR signaling. EGFR is ubiquitously expressed in the epithelial layers of the lung, gut, and skin, and is related to several cancers that show overexpression and mutations in EGFR, such as non-small cell lung cancer (NSCLC), head and neck squamous cell carcinoma (HNSCC), melanoma, breast and ovarian cancers, and glioblastoma (GBM) [44,45,46,47,48]. EGFR also plays key roles in homeostatic regulation of cellular proliferation, inhibition of apoptosis, increased cell migration and differentiation, mucus production, activation of the inflammatory response, and maintenance of cell survival [49]. EGFR is located on the plasma membrane and is widely expressed in other intracellular organelles such as endosomes, mitochondria, the nucleus, and lysosomes [50].

## 3. Activation and Regulation of EGFR Signaling

EGFR activation induces dimerization and phosphorylation of the molecule, in which tyrosine-phosphorylated sites promote the activation of downstream signaling cascades and EGFR internalization. In downstream signaling, phosphorylated tyrosine residues of EGFR recruit specific adapters and activate effector proteins to trigger certain signaling molecules such as mitogen-activated protein kinases (MAPKs), Janus kinase (JAK)/signal transducer and activator of transcription (STAT), phospholipase C-gamma 1 (PLCγ1), and phosphatidylinositide 3-kinases (PI3Ks). Phosphorylation of tyrosine (Y) residues 1068, Y1086, Y1101, Y1148, Y703, Y845, and Y974 is responsible for the activation of EGFR functions [51,52]. During EGFR internalization, EGFR moves to endosomes and is subsequently recycled to the cell surface or trafficked to other intracellular organelles and the nucleus [53]. EGFR internalization also induces EGFR endocytosis in a clathrin-dependent or clathrin-independent manner [54,55]. The nuclear portion of EGFR is translocated with importin β1, which has a nuclear localization sequence (NLS) to activate DNA-dependent protein kinase or regulate the transcriptional activity of many important genes for cell proliferation and survival [54,55,56]. EGFR is also translocated into the mitochondria, which is the main source of ATP and reactive oxygen species generation, and subsequently regulates mitochondrial bioenergetics [57]. The overall EGFR signaling network orchestrates cell survival to control cellular homeostasis.

## 4. EGFR as a Receptor for Viral Entry

Viral infections are a recurring challenge to human health, with various viruses employing diverse strategies to invade host cells. Understanding the mechanisms underlying viral entry is essential for developing effective antiviral strategies. In the following section, we discuss the critical role of EGFR in the entry of both DNA and RNA viruses (Figure 1A).

### 4.1. DNA Viruses

Human cytomegalovirus (HCMV) causes severe and fatal diseases in immunocompromised individuals and is associated with atherosclerosis, coronary restenosis, and virus-induced birth defects. HCMV is an enveloped, linear, double-stranded DNA virus [58] that uses EGFR as a receptor to enter host cells and induce critical downstream steps in the viral life cycle. The ErbB family cDNAs transfected into Chinese hamster ovary cells shows critical binding of the HCMV envelope glycoprotein B (gB) to EGFR [9]. HCMV also establishes a latent infection and persistently infects myeloid-lineage cells [59]. CD34+ human progenitor cells (HPCs) are replication-restricted cells that are crucial reservoirs of HCMV latency [60]. EGFR signaling contributes to the successful establishment of viral latency during early events in CD34+ cells. HCMV entry, cellular trafficking, and nuclear translocation are mediated by the activation of downstream EGFR signaling in HPCs. The EGFR inhibitor AG1478 upregulates the expression of HCMV lytic IE1/IE2 mRNA, downregulates latency-associated UL138 mRNA, and alters the expression of cellular hematopoietic interleukin 12 (IL-12). EGFR also acts as a determinant in the selection of hematopoietic cells for HCMV latency tropism and is important in the early stages of successful HCMV infection [10].

Hepatitis B virus (HBV) is an enveloped virus with a circular DNA genome, and sodium taurocholate co-transporting polypeptide (NTCP) is required for HBV entry into cells [61,62]. Among the factors determining the susceptibility of cells to HBV infection, EGFR plays a critical role in inducing the internalization of HBV virions. Molecular interactions between NTCP and EGFR are important for supporting viral infection, and point mutations in NTCP and inactivation of EGFR can disrupt the NTCP–EGFR interaction. On the host cell surface, HBV attaches to EGFR-NTCP to promote the internalization of HBV virions that cross the plasma membrane of cells [11].

### 4.2. RNA Viruses

Influenza A viruses (IAVs) are enveloped viruses with single-stranded, negative-sense RNA genomes. Sialic acids are receptors for IAVs on cellular surfaces and facilitate the entry of viral particles via endocytic pathways [63]. EGFR also facilitates the uptake of IAVs through multivalent IAV binding to generate clusters of lipid rafts and activate EGFR signaling and facilitate IAV entry [12].

Hepatitis E virus (HEV) is a non-enveloped virus with a single-stranded, positive-sense RNA genome that causes acute viral hepatitis worldwide [64]. EGFR serves as a novel host receptor for HEV entry without affecting the viral RNA replication process to propagate the virus. Ectopic expression of EGFR shows a proviral role in supporting HEV entry into HepaRG cells and primary human hepatocytes [13].

Hepatitis C virus (HCV) is an enveloped virus with a single-stranded, positive-sense RNA genome that causes acute and chronic hepatitis related to hepatocellular carcinoma (HCC) and death [65,66]. A functional RNAi kinase screening assay identified EGFR and ephrin receptor A2 as the host co-factors for HCV infection. EGFR kinase inhibitors have shown antiviral activity against HCV infection in both cell culture and animal model systems [14]. The interaction between HCV and Cd81 induces EGFR activation. The internalization of EGFR facilitates the entry of HCV, which is enhanced by EGFR ligands. EGFR inhibition suppresses EGFR-mediated endocytosis during HCV infection [15].

## 5. EGFR Trafficking for Proviral Functions

Understanding the complex interplay between viruses and EGFR in host cells is paramount for devising effective strategies to combat infectious diseases. In the following section, we discuss the role of EGFR as a versatile player in the replication of both DNA and RNA viruses through its translocation into intracellular organelles (Figure 1B,C).

### 5.1. DNA Viruses

Epstein–Barr virus (EBV) infection is associated with multiple malignancies, including Hodgkin’s disease, Burkett’s lymphoma, and nasopharyngeal carcinoma (NPC) [67]. EBV is an enveloped double-stranded DNA virus (dsDNA) [68], while EBV latent membrane protein 1 (LMP1) is an oncoprotein related to EBV-induced cell transformation that activates several signaling pathways, such as the phosphatidylinositol 3-kinase (PI3K)/Akt and NF-κB signaling pathways [69,70,71,72]. Carboxyl-terminal activation regions 1 and 2 (CTAR1 and CTAR2) are two major signaling domains of LMP1 that are associated with the activation of the NFκB signaling pathway by the binding of tumor necrosis factor receptor-associated factors (TRAFs) [73,74,75]. An NPC cell line showed increased EGFR expression, which was related to the level of LMP1 [76,77]. The CTAR1 domain of LMP1 is essential for activation of the EGFR promoter. Two proximal nuclear factor (NF)-κB binding sites on the EGFR promoter co-operate to promote EGFR transcriptional activity, which is regulated by LMP1 [78]. LMP1-CTAR1 also promotes Bcl-3 mRNA expression, the translocation of Bcl-3/p50 into the nucleus, and the activation of the signal transducer and activator of transcription 3 (STAT3). Subsequently, the Bcl-3 and p50 complexes activate EGFR expression in the nucleus. The induction of Bcl-3 and EGFR expression is regulated by LMP1-CTAR1, which is suppressed by STAT3 inhibitors [79]. PKCδ acts as a key mediator for the activation of STAT3 and EGFR by LMP1-CTAR1 [16]. In EBV infection, intracellular EGFR accumulates in the nucleus, where it binds to the cyclin D1 promoter by regulating LMP1 in NPC carcinogenesis [50,80]. LMP also mediates the accumulation of EGFR and STAT3 interactions and enhances the activation of cyclin D1 promoter in the nucleus [17].

Vaccinia virus (VACV) is an enveloped virus with a linear, double-stranded DNA genome. VACV has been successfully used as a vaccine tool to eradicate smallpox. Innate and adaptive immune responses are suppressed in dendritic cell infections with VACV [81,82]. The mitochondria-targeted viral protein F1L is a Bcl-2 homolog that suppresses apoptosis. To induce cell survival and viral spread, vaccinia growth factor (VGF), a homolog of the epidermal growth factor, is expressed and activates EGFR phosphorylation during VACV infection [18,83,84]. Upon VACV infection, VGF-induced EGFR activation and F1L co-operate to maintain cell survival against infection-induced apoptotic activity [19]. F11L is a viral protein that inhibits RhoA signaling and regulates cell motility. EGFR and VGF co-operate the regulation of cell motility together with F11L to efficiently promote the cell-to-cell spread of VACV infection [20,85].

Human papillomavirus (HPV) infection is known to increase the risk of cervical and oropharyngeal cancers. HPV is a non-enveloped virus with a double-stranded genomic DNA. HPV E7 is a major oncoprotein that is highly expressed in HPV-induced malignancies [86,87,88]. HPV E7 targets the AP-2 (clathrin-adapter protein 2) complex and contains specific AP2 recognition sequences. During clathrin-mediated endocytosis, EGFR induces the internalization of HPV E7, which contributes to cellular transformation by regulating protein trafficking [89].

### 5.2. RNA Viruses

Understanding the host response to RNA virus infection includes recognizing the importance of RNA degradation pathways in the antiviral innate immune response, especially the inhibition of viral replication, including viral RNA degradation. RNA viruses are identified by RIG-I-like receptors, namely, RIG-I and MDA5, that trigger the production of type I and III interferons. The DExD/H box (DDX) 60 helicase plays an essential role in RIG signaling to induce IFN production in response to viral RNA [90,91,92]. Within the system that recognizes viral RNA, virus-mediated EGFR activation is essential for promoting DDX60 phosphorylation, which, in turn, mitigates the DDX60-mediated antiviral response [93].

The phosphorylation of EGFR and activation of extracellular signal-regulated kinase (ERK) occur at an early stage of IAV infection. The upregulation of type I and type III interferons (IFNs) and interferon-stimulated genes (ISGs) is induced by the inhibition of EGFR or ERK, with suppression of viral replication during IAV infection. Activation of EGFR/ERK signaling supports an IAV-induced immune evasion mechanism against host antiviral innate immunity. The Src homology region 2 containing protein tyrosine phosphatase 2 (SHP2) is also activated by IAV infection and is related to the proviral functions of EGFR signaling both in vitro and in vivo. During IAV infection, SHP2 is a major regulator of EGFR/ERK signaling for immune evasion, which facilitates viral replication and virion production [21]. IAV infections are highly associated with acute respiratory illnesses. Mucins are expressed within the mucus and play a critical role in lung defense against respiratory pathogens such as IAV [94,95]. EGFR and EGFR phosphorylation are involved in IAV-mediated MUC5AC production. The secretion of MUC5AC occurs on the surface of epithelial cells both in vitro and in vivo and is upregulated by IAV infection, which is dependent on the EGFR–ERK–specificity protein 1 (Sp1) signaling pathway to maintain lung homeostasis [22].

Respiratory syncytial virus (RSV) causes lower respiratory tract infections in children and exacerbates chronic lung diseases in adults. RSV is an enveloped virus with a single-stranded, negative-sense RNA genome [96,97]. Upon RSV infection, activation of ERK signaling induces an inflammatory response and maintains the survival of virus-infected cells [98]. Increased EGFR activity is involved in the protection of lung epithelial cells against RSV infection. RSV induces EGFR activation in response to IL-8 and ERK secretion. RSV also coordinates the balance of Bcl2 protein with increased BclxL and decreased BimEL levels to prolong cell survival by activating EGFR [24]. In airway epithelial antiviral responses, CXCL10 promotes the recruitment of lymphocytes to respiratory pathogen-infected cells. Upon respiratory viral infection, EGFR is activated and subsequently suppresses CXCL10 production, which is dependent on IFN regulatory factor (IRF) 1, and the inhibition of EGFR increases IRF1 and CXCL10 levels [23]. For the suppression of airway epithelial antiviral immune responses such as the production of IFN-λ, RSV induces EGFR activation and promotes viral replication, which is mediated by the RSV F protein. During RSV infection, the upregulation of IRF1 and IFN-λ and suppression of RSV production are involved in EGFR inhibition [25]. Clinically isolated RSV A2001/2-20 (2-20), which is a clinically relevant phenotype, has been shown to cause increased airway mucin expression in response to RSV infection [99]. The activation of phosphorylated EGFR is increased during RSV 2-20 infection, and MUC5AC expression is decreased by EGFR inhibitors both in vivo and in vitro [26]. Treatment with the antioxidant *N*-acetyl-l-cysteine (NAC), an anti-inflammatory and antioxidant agent, has been shown to suppress the activation of EGFR and MUC5AC against RSV infection in BEAS-2B cells [100].

Rhinoviruses (RVs) are non-enveloped viruses with a positive-stranded RNA genome. RVs cause common colds and are associated with asthma and chronic obstructive pulmonary disease. RV infections mediate overproduction of mucus, which promotes pathogenesis and disease exacerbation [101,102,103]. RV-induced mucin expression is necessary to promote viral replication in primary human epithelial cells, and its mechanism depends on Toll-like receptor 3 (TLR3) signaling. The expression of EGFR ligands, such as TGF-α and amphiregulin, activates TLR3-mediated EGFR/ERK signaling and mucin expression by inducing an autocrine/paracrine cycle [27]. MUC5AC is the major mucin in bronchial epithelial cells and is upregulated during RV infection [104]. Activation of the EGFR/NF-κB pathways promotes Sp-1 transactivation of the MUC5AC promoter to increase RV-mediated MUC5AC production [28]. RV infections induce abnormal neutrophilic inflammation, which exacerbates RV-mediated inflammatory lung disease [105]. The upregulation of EGFR is also related to neutrophil infiltration, which is activated by interleukin (IL)-8 and intercellular adhesion molecule-1 (ICAM-1) in RV-infected bronchial epithelial cells. EGFR inhibitors suppress the expression of IL-8 and ICAM-1 [29].

Infection with the Japanese encephalitis virus (JEV) induces viral encephalitis by disrupting the blood–brain barrier (BBB), the first immune barrier against pathogens. JEV is an enveloped virus with a single-stranded, positive-sense RNA genome [106,107]. In human brain microvascular endothelial cells (hBMECs) and mouse brains infected with JEV, EGFR signaling was shown to be activated to promote immune evasion mechanisms. EGFR phosphorylation is activated at an early stage after JEV infection, while propagation of viral particles has been shown to be suppressed in an EGFR-KO hBMEC cell line or by treatment with EGFR inhibitors. EGFR-KO hBMECs upregulate IFNs and downregulate viral production [30].

Severe acute respiratory syndrome coronavirus (SARS-CoV) is associated with severe respiratory syndromes such as coronavirus disease 2019 (COVID-19) and SARS-CoV-1-mediated acute lung injury [108,109]. EGFR activation controls the wound-repair pathways that are essential for respiratory virus-induced lung injury. During SARS-CoV-1 infection, the expression of EGFR ligands increases, accelerating lung disease and affecting wound-repair mechanisms [31]. In SARS-CoV-2-infected human bronchial epithelial (HBE) cells, increased expression of MUC5B/MUC5AC, inflammatory cytokines, and EGFR ligands is related to mucin gene regulation. The inhibition of EGFR pathways suppresses mucin hypersecretion during SARS-CoV-2 infection [32].

Dengue virus (DENV) and Zika virus (ZIKV) are flaviviruses, which are enveloped viruses with a single-stranded, positive-sense RNA genome and are associated with mosquito-borne viral diseases worldwide [110]. Tyrosine kinase activity and signal transduction are involved in flavivirus replication [111]. EGFR is involved in RTK activity for the replication of flaviviruses such as DENV and ZIKV [33,34]. Monocytes and macrophages upregulate cytokine production and viral replication during the early stages of DENV infection. Inhibition of EGFR and NF-kB induces downregulation of antiviral cytokine production and DENV replication [33].

## 6. Opposing Regulation by EGFR to Control Viral Infection

EGFR plays a pivotal role in the intricate interplay between viruses and host cells. It plays a dual, multifaceted role, functioning as both an enabler and a suppressor of viral infection, and its activity is often contingent on the specific viral state. These intricate functions rely on the precisely orchestrated regulation of EGFR activation, which can effectively toggle between various viral infection states, such as latency and reactivation, or profoundly affect discrete phases of the viral life cycle.

### 6.1. Opposite Roles of EGFR in Latency and Reactivation of Viral Infection

EGFR mediates opposing regulatory mechanisms to switch between the viral reactivation and the latent states [112]. HCMV latency and reactivation in CD34+ hematopoietic progenitor cells are regulated by HCMV microRNAs (miRNAs), small RNAs that decrease protein expression [113]. HCMV miR-US5-2 downregulates GAB1, an adapter protein that regulates EGFR-induced MEK/ERK signaling. The transcription factor early growth response gene 1 (EGR1), which is located downstream of EGFR-induced MEK/ERK signaling, also regulates the expression of the HCMV latency determinant UL138. HCMV miR-US5-2 is a critical modulator that targets GAB1, attenuates EGFR signaling pathways, and interferes with EGR1 and HCMV UL138 expression during HCMV infection [114]. Ultimately, attenuation of EGFR signaling induces HCMV reactivation from latency. HCMV coordinates with two viral genes, UL135 and UL138, to reactivate the lytic cycle from a latent state. HCMV UL135 is a critical factor in HCMV reactivation against the suppressive effector HCMV UL138, which performs the opposite function in regulating viral replication. UL135 and UL138 of HCMV target the same receptor, EGFR, to regulate the viral life cycle. Thus, HCMV UL138 induces EGFR expression and activation of the cell surface, while HCMV UL135 reduces EGFR expression on the host cell surface for reactivation. Inhibition of EGFR induces reactivation from latency and viral replication. Thus, the combination of UL135 or UL135 with EGFR represents a molecular switch that regulates the latency or reactivation of HCMV infection [35,112]. HCMV UL135 interacts with Src homology 3 (SH3) domain-containing kinase-binding protein 1 (SH3KBP1) and Abelson-interacting protein-1 (Abi-1), which have SH3 domains as host adapter proteins, and these protein complexes are required for regulating EGFR signaling in HCMV reactivation [115].

### 6.2. Opposite Roles of EGFR at Different Stages of Viral Infection

Activation of EGFR signaling occurs via virus binding and is immediately inhibited in the early stages of infection. HBV is an enveloped virus with a circular DNA genome, and NTCP is required for HBV entry [61,62]. In assessments of the susceptibility of cells to HBV infection, EGFR has been shown to be critical for inducing the internalization of HBV virions. Molecular interactions between NTCP and EGFR are important for supporting viral infection. Point mutations in NTCP and inactivation of EGFR disrupt the NTCP-EGFR interaction. On the host cell surface, HBV induces attachment to EGFR-NTCP to promote the internalization of HBV virions that cross the plasma membrane of cells [11]. Moreover, NTCP serves as a receptor for HBV entry, and EGFR is also involved in NTCP-mediated entry. Specifically, the EGFR endocytosis machinery is required for the internalization process in HBV entry, which involves the phosphorylation of EGFR and recruitment of adapter-related protein complex 2 subunit alpha 1 (AP2A1) and EGFR pathway substrate 15 (EPS15) as adapter molecules. The factors responsible for EGFR activation, such as the EGFR-sorting machinery, are involved in EGFR ubiquitination, such as the signal-transducing adapter molecule (STAM) and lysosome-associated protein transmembrane 4 beta (LAPTM4B), and promote the localization of HBV preS1 along with EGFR transport from endosomes to lysosomes. In late endosomes, EGFR transport is essential for promoting productive HBV infections [38]. In contrast, after HBV infection, such as the stages after viral entry, EGFR is one of the major targets of HBV-encoded X (HBx) to control cell growth. HCC is associated with HBV infection through the expression of the HBx protein to regulate hepatocarcinogenesis. HBx increases miR-7 expression, which targets EGFR mRNA to decrease the expression of EGFR and induces slow cell growth in HCC. HBx-miR-7-EGFR regulation is critical for controlling the growth rate of HCC cells [39].

Herpes simplex virus type 1 (HSV-1) is a common pathogen causing cold sores that progress to herpes keratitis and herpes simplex encephalitis, and also causes serious disease in transplant recipients. HSV-1 is an enveloped virus that contains double-stranded DNA. HSV-1 establishes latency in sensory neurons and can be reactivated to produce progeny virions in epithelial cells [116,117]. For HSV-1 infection into neuronal cells, reorganization of the actin cytoskeleton is essential to promote the entry of HSV-1 into neuronal cells. F-actin and cofilin regulate the efficacy of HSV-1 entry. The initial activation of the EGFR signaling pathway by binding to HSV-1 induces F-actin polymerization and cofilin phosphorylation. EGFR suppresses the infectivity of HSV-1 without affecting viral binding to the cells [40]. In contrast, infected cell protein 0 (ICP0) of HSV-1 encodes an SH3 domain-binding site, which is essential for the downregulation of EGFR. Both the surface levels and total EGFR expression decrease during HSV-1 infection [41].

EGFR is also downregulated in the early stages of human adenovirus (HAdV) infection. Specifically, E3 of human group C adenoviruses contributes to EGFR downregulation [118,119]. HAdV is a non-enveloped virus with linear double-stranded DNA [120]. In adenovirus pathogenesis, the early transcription region 3 (E3) of the adenovirus, E3-13.7, is an integral membrane protein that associates with EGFR to alter EGFR trafficking. The residues 675–697 of EGFR include lysosomal sorting signals, which are required for E3-13.7-induced downregulation in early endosomes. E3-13.7 and the EGFR complex are located in the early endocytic compartments and then dissociate, promoting the recycling of EGFR and the retention of E3-13.7 in endosomes [121]. Adenovirus infection triggers the stress-induced EGFR trafficking pathway, which activates the host innate immune response. The E3 RIDα protein is encoded by group C adenoviruses, which promotes the downregulation of the EGFR/NFκB signaling pathway. Interestingly, stress-induced pathways of EGFR trafficking are related to severe disease, which depend on specific adenovirus serotypes without conserved RIDα [122].

The expression and functionality of EGFR have also been determined in monocytic leukemic cell lines and macrophage subpopulations [123]. HCMV infects peripheral blood monocytes, which mediate the transfer of virus particles to latency sites such as the bone marrow. Monocytes are critical for understanding HCMV pathogenesis in the host. For productive HCMV propagation, the ability of the virus to cross the cell membrane and translocate viral DNA into the nucleus is crucial for replication of the HCMV genome [124]. HCMV infection promotes the activation of EGFR signaling, which is required for entry into monocytes and stimulates cell movement. The actin nucleator neural Wiskott–Aldrich syndrome protein (N-WASP) normally controls actin growth in leukocytes. However, upon viral infection, activated EGFR induces the stimulation of highly activated N-WASP to promote cell movement [125,126], while N-WASP knockdown inhibits HCMV-induced monocyte motility. The inhibition of EGFR activation can suppress viral entry into monocytes, and EGFR activation plays a key role in mediating viral entry into monocytes and promoting viral spread during HCMV pathogenesis [36]. In the post-entry steps of HCMV infections, EGFR kinase activity regulates the subcellular localization of the viral particles in monocytes. Activation of EGFR signaling by inducing HCMV gB-EGFR internalization is required for the translocation of viral DNA into the host nucleus and productive HCMV infection [37]. However, the role of EGFR in HCMV infection is controversial and depends on the virus strain or cell line specificity [9,127,128]. Downregulation of EGFR expression is a characteristic of HCMV infection. HCMV early gene products are necessary to suppress EGFR expression [128]. HCMV infection mediates the upregulation of Wilms’ tumor 1 (WT1) protein, a transcription factor associated with the negative regulation of EGFR. The binding of WT1 to the EGFR promoter also increases during HCMV infection, while depletion of WT1 suppresses HCMV-induced negative regulation of EGFR. HCMV-induced WT1 acts as a negative regulator to maintain the reduction in EGFR mRNA levels [129].

## 7. Discussion

We assessed the existing knowledge regarding the interactions between EGFR and various viruses, which play a crucial role in the efficient propagation of these viruses. Additionally, we curated well-defined data regarding the relationships of multiple viral species with EGFR, as depicted in Figure 1, and proposed a schematic representation to elucidate the role of EGFR in viral infections. As summarized in this review, both DNA and RNA viruses harness the EGFR protein in a proviral capacity. For example, the Zaire Ebola virus (ZEBOV) activates the PI3K pathway to facilitate its entry into host cells. ZEBOV is notorious for inducing severe hemorrhagic fever with a 90% mortality rate in humans, representing a formidable public health concern. Thus, these observations underscore the potentially pivotal role of EGFR in severe infections by viruses such as ZEBOV [114,130,131]. The significance of EGFR dynamics is not limited to human viruses but extends to animal viruses as well. Two examples of animal viruses, namely, Bovine Parainfluenza Virus 3 (BPIV3) and Transmissible Gastroenteritis Virus (TGEV), also utilize the EGFR signaling pathways for host cell entry. BPIV3 is a notable respiratory virus in cattle, and its entry into Madin–Darby Bovine kidney (MDBK) cells is mediated by the clathrin-dependent endocytosis pathway, which can be inhibited by an EGFR inhibitor. Activation of EGFR signaling leads to an increase in viral infectivity during the entry process. Furthermore, EGFR inhibition represses the rearrangement of the F-actin cytoskeleton. TGEV is associated with severe diarrhea in newborn piglets, leading to high mortality rates. EGFR interacts with the spike protein of TGEV, facilitating the formation of the EGFR and aminopeptidase N (the receptor for TGEV) complex, which accelerates TGEV entry into host cells [132,133]. Having explored the pivotal role of EGFR in diverse viral infections, we suggest that the growing emphasis on its significance positions EGFR as a promising candidate for future antiviral development.

## Figures and Tables

**Figure 1 biomolecules-13-01766-f001:**
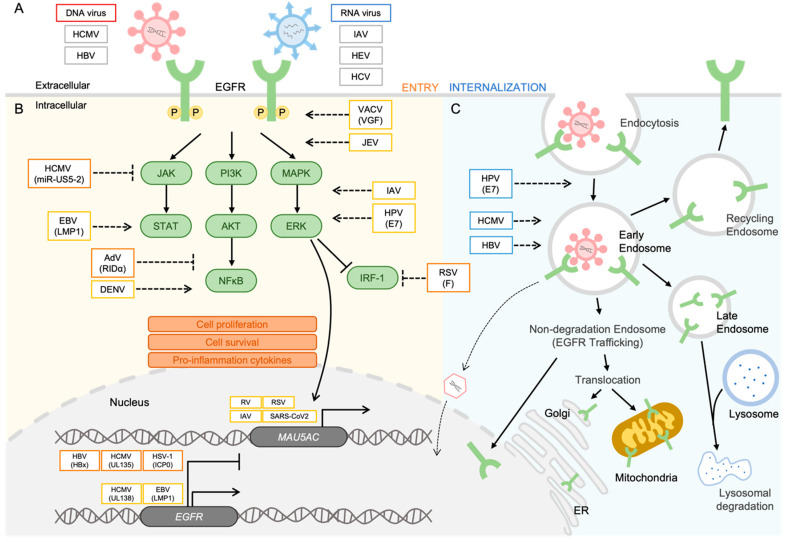
Viruses modulate the EGFR signaling network. (**A**) For entry, gB of HCMV binds to EGFR and induces internalization of EGFR and virus particles. HBV-NTCB interacts with EGFR, facilitating the entry process. EGFR acts as a co-factor on the plasma membrane to facilitate host cell entry in IAV, HEV, and HCV infections. (**B**) For regulation of the EGFR signaling pathway and EGFR expression, accumulation of EGFR is observed in the nucleus. UL138 of HCMV and LMP1 of EBV induce upregulation of EGFR expression. Conversely, HBx of HBV, UL135 of HCMV, and ICP0 of HSV-1 suppress EGFR production. The expression of MUC5AC is increased by respiratory virus infections such as RV, SARS-CoV-2, IAV, and RSV. The phosphorylation of EGFR is triggered by VACV and JEV infection, and the MAPK/ERK pathway is activated by IAV and HPV-6 infection. IRF-1 is suppressed in RSV infection. In the PI3K/Akt pathway, DENV induces NF-κB activation, whereas the RIDα of AdV inhibits NF-κB. The miR-US5-2 of HCMV inhibits the activation of the Jak/STAT pathway, whereas LMP1 of EBV upregulates STAT activation. (**C**) For EGFR trafficking with viral proteins into intracellular organelles, E7 of HPV promotes the internalization of EGFR, and EGFR trafficking into endosomes is essential for HCMV and HBV replication. This figure was generated by using a drawing tool available in Microsoft PowerPoint version 2023.

**Table 1 biomolecules-13-01766-t001:** Role of EGFR in viral infection.

Virus genome	Virus name	Animal/Cells	EGFR inhibitors		Molecular mechanism	References
			EGFR TKI ^a^	EGFR Ab ^b^		
EGFR as receptor for viral entry (Extracellular)						
DNA	HCMV	HEL, CD34+ HPCs, MB453, MB468 cells	AG1478, Gefitinib		Binding of gB-EGFR	[9,10]

	HBV	PHHs, dHepaRG cells	Gefitinib		Interaction of NTCP-EGFR	[11]

RNA	IAV	A549 cells	Gefitinib		Activation of lipid-rafts and EGFR signaling	[12]

	HEV	HepG2 cells	Erlotinib	Cetuximab	Binding of HEV-EGFR	[13]

	HCV	Huh7.5.1, PHH cells	Lapatinib	Cetuximab	Cofactors for entry	[14,15]
		and Chimeric uPA/ SCID mice	Erlotinib		Internalization of EGFR	
Proviral role of EGFR (Intracellular)						
DNA	EBV	C33A, CNE1 cells	AG1478		Induction of EGFR promoter by LMP1	[16,17]
					Accumulation of EGFR and STAT3 in nucleus	

	VACV	Hep2, HELA, BSC40 cells	AG1478, PD153035, Vandetanib, Gefitinib		Activation of EGFR phosphorylation by VGF Regulation of cell motility	[18,19,20]

RNA	IAV	A549, NCI-H292 cells and C57BL/6 mice	Afatinib, PD168393, Gefitinib		Activation of SHP2 Upregulation of MUC5AC production	[21,22,23]


	RSV	H292, A549, hTBEs, NHBE, BEAS-2b cells and BALB/cJ mice	AG1478, PD153035, Erlotinib		Suppression of IRF1 Suppression of CXCL10 Upregulation of MUC5AC production	[23,24,25,26]
	RV	NCI-H292, BEAS-2B cells	AG1478		Upregulation of MUC5AC production	[27,28,29]
					Induction of IL-8 and ICM-1	

	JEV	hBMECs, BHK-21 cells	AG1478, Gefitinib		Promotion of immune evasion	[30]

	SARS-CoV	HBE cells	Gefitinib		Upregulation of EGFR ligands	[31,32]
					Upregulation of MUC5AC/MUC5B	

	DENV	Monocytes, HEK-293 cells	Gefitinib, Afatinib		Upregulation of cytokine production	[33,34]

	ZIKV	hBMECs	AG1478		Activation of HER2	[30]
Dual role of EGFR						

DNA	HCMV	Monocytes, CD34+ HPC cells	AG1478, Gefitinib,		Induction of EGFR expression by UL138	[35,36,37]
			Erlotinib		Suppression of EGFR expression by UL135	
					Regulation of viral particle trafficking	

	HBV	HepG2-NTCP cells	Gefitinib		Translocation of EGFR into endosome	[38,39]
					Suppression of EGFR expression by HBx	

	HSV-1	SK–N–SH cells	AG1478		Regulation of cofilin activity	[40,41]
					Suppression of EGFR expression by ICP0	

^a^ EGFR TKI: EGFR tyrosine kinase inhibitor. ^b^ EGFR Ab: EGFR monoclonal antibody to block extracellular ligand binding domain. Abbreviations: gB, HCMV glycoprotein B; HPCs, human progenitor cells; HEL, human erythroleukemia; MB468/ MB453 cells, human breast cancer cells; HEK-293T, human embryonic kidney cells; PHHs, primary human hepatocytes; dHepaRG, differentiated human hepatocyte carcinoma cell line; A549 cells, adenocarcinomic human alveolar basal epithelial cells; HepG2 cells, laryngeal carcinoma cells; Huh7.5.1 cells, human hepatoma-derived cells; Chimeric uPA/SCID mice, albumin enhancer/promoter-driven urokinase-type plasminogen activator/severe combined immunodeficient mice; CNE1, LMP1-negtive poorly differentiated nasopharyngeal carcinoma cell line; C33A, LMP1-CTAR1-expressing cells; BSC40, African green monkey kidney cells; NHBE, normal human bronchial epithelial cells; NCI-H292, human pulmonary mucoepidermoid carcinoma cell line; BEAS-2b, human normal bronchial epithelial; hTBEs, human tracheobronchial epithelial cells; HBE, human bronchial epithelial cells; HepG2-NTCP, HepG2 cells expressing NTCP; hBMECs, human brain microvascular endothelial cells; SK–N–SH cells, human neuroblastoma cell line.

## Data Availability

Not applicable.

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
