# Peer review of "Role of Virus-Induced EGFR Trafficking in Proviral Functions"

_biomolecules, 2023, doi:10.3390/biom13121766_

Round 1
Reviewer 1 Report
Comments and Suggestions for Authors
The interplay between viral infections and EGFR signaling has been widely recognized. Numerous reviews have also explored EGFR's intracellular role in viral infection and pathogenesis. This article focuses on the proviral function of EGFR trafficking. The entire manuscript is lucidly structured and succinctly captures the up-to-date information. Nevertheless, certain concerns need attention.
1, the reference number in table 1 is incorrect entirely.
2, The potential effects of EGFR translocation to Golgi/mitochondria or EGFR autophagic degradation on viral infection should be illustrated.
3, Whether EGFR regulates innate antiviral immunity and its impact on viral infection.
4, Have EGFR inhibitors been clinically utilized in the treatment of infectious diseases?
Comments on the Quality of English LanguageModerate editing of English language required
Author Response
Dear Reviewer,
Thank you for your review of our manuscript. We have carefully corrected the issues raised by the reviewer and revised our manuscript accordingly.
Please find our detailed responses shown in blue. The revised parts in the manuscript are highlighted using a yellow marker. Please see the attachment.
We believe that our revisions address the reviewers’ concerns and we look forward to the publication of our manuscript in Biomolecules. Thank you very much for your help and attention.
Sincerely Yours,
Hye Jin Shin, Ph. D.
<Reviewer 1>
The interplay between viral infections and EGFR signaling has been widely recognized. Numerous reviews have also explored EGFR's intracellular role in viral infection and pathogenesis. This article focuses on the proviral function of EGFR trafficking. The entire manuscript is lucidly structured and succinctly captures the up-to-date information. Nevertheless, certain concerns need attention.
Response: We thank the reviewer for the positive and constructive comments.
1, The reference number in table 1 is incorrect entirely.
Response 1. Thank the reviewer for finding the error. References for Table 1 were corrected with appropriate references
2, The potential effects of EGFR translocation to Golgi/mitochondria or EGFR autophagic degradation on viral infection should be illustrated.
Response 2. In accordance with the published papers, the Epidermal Growth Factor Receptor (EGFR) undergoes translocation, exerting a significant influence within intracellular organelles such as the Golgi apparatus or mitochondria. Nevertheless, the precise functional ramifications of EGFR translocation within these intracellular organelles in the context of viral infection remain to be fully elucidated. In this review paper (Figure 1), we illustrated only the role of EGFR in viral infection as directly defined by experiments.
3, Whether EGFR regulates innate antiviral immunity and its impact on viral infection.
Response 3. We briefly summarized the role of EGFR in innate antiviral immunity, such as interferon expression, against certain virus infections in Section 5.
4, Have EGFR inhibitors been clinically utilized in the treatment of infectious diseases?
Response 4. Unfortunately, the clinical utilization of EGFR inhibitors as antiviral agents for treating infectious diseases remains unrealized.
Reviewer 2 Report
Comments and Suggestions for Authors
In the current study, authors presented a comprehensive summary of the current state of knowledge regarding the intricate interactions between EGFR and viruses. This study is interesting and well-presented that it can be accepted after minor revision.
Here are some points:
· In Table 1, “the receptor role of EGFR (extracellular)” must be omitted or the place of it must be changed.
· There are gliding problems in Table 1. The alignment must be rearranged.
· There are a lot references in relation of EGFR with several cancers. Authors only mentioned one reference, reference 11.
· Authors should give reference which drawing tool that they used for Figure 1.
· Authors could explain the RNA viruses initially in Section 5.2.
· Opposite roles of EGFR could be also divided for RNA and DNA viruses.
· Authors must explain the roles of EGFR inhibitors that they mentioned in Table 1 such as how they show their effects on EGFR. They could also insert a Fifure of the chemical structures of these inhibitors.
Author Response
Dear Reviewer,
Thank you for your review of our manuscript. We have carefully corrected the issues raised by the reviewer and revised our manuscript accordingly.
Please find our detailed responses shown in blue. The revised parts in the manuscript are highlighted using a yellow marker. Please see the attachment.
We believe that our revisions address the reviewers’ concerns and we look forward to the publication of our manuscript in Biomolecules. Thank you very much for your help and attention.
Sincerely Yours,
Hye Jin Shin, Ph. D.
<Reviewer 2>
In the current study, authors presented a comprehensive summary of the current state of knowledge regarding the intricate interactions between EGFR and viruses. This study is interesting and well-presented that it can be accepted after minor revision.
Response: We thank the reviewer for the positive and constructive comments.
Here are some points:
Comments 1. In Table 1, “the receptor role of EGFR (extracellular)” must be omitted or the place of it must be changed.
Response 1. Thank you for pointing this out. We omitted “the receptor role of EGFR” in Table 1.
Comments 2. There are gliding problems in Table 1. The alignment must be rearranged.
Response 2. We have now modified the table 1 as suggested.
Comments 3. There are a lot references in relation of EGFR with several cancers. Authors only mentioned one reference, reference 11.
Response 3. We fully agree with the reviewer’s comment. In accordance with the reviewer’s suggestion, we added references in our revised manuscript.
Comments 4. Authors should give reference which drawing tool that they used for Figure 1.
Response 4. We created Figure 1 by using the drawing tool available in Microsoft PowerPoint version 2023. We added this sentence in Figure legend.
Comments 5. Authors could explain the RNA viruses initially in Section 5.2.
Response 5. As the reviewer suggested, we briefly summarized the RNA viruses initially in Section 5.2.
Comments 6. Opposite roles of EGFR could be also divided for RNA and DNA viruses.
Response 6. In section 6. We distinguish the opposite roles of EGFR based on characteristics of the virus infection state such as latency or early stage after entry of the virus. Most DNA viruses are related to the opposite roles of EGFR and are summarized in this review paper.
Comments 7. Authors must explain the roles of EGFR inhibitors that they mentioned in Table 1 such as how they show their effects on EGFR. They could also insert a Fifure of the chemical structures of these inhibitors.
Response 7. As the reviewer commented, we explained the targets of EGFR inhibitors in Table 1.